# Tumor Suppressive Circular RNA-102450: Development of a Novel Diagnostic Procedure for Lymph Node Metastasis from Oral Cancer

**DOI:** 10.3390/cancers13225708

**Published:** 2021-11-15

**Authors:** Toshiaki Ando, Atsushi Kasamatsu, Kohei Kawasaki, Kazuya Hiroshima, Reo Fukushima, Manabu Iyoda, Dai Nakashima, Yosuke Endo-Sakamoto, Katsuhiro Uzawa

**Affiliations:** 1Department of Oral Science, Graduate School of Medicine, Chiba University, Chiba 260-8670, Japan; axga0845@chiba-u.jp (T.A.); afda8097@chiba-u.jp (K.K.); kazuhiro06281988@gmail.com (K.H.); nakashimad@chiba-u.jp (D.N.); 2Department of Dentistry and Oral-Maxillofacial Surgery, Chiba University Hospital, Chiba 260-8677, Japan; fukushimar@chiba-u.jp (R.F.); iyodam@chiba-u.jp (M.I.); end@faculty.chiba-u.jp (Y.E.-S.)

**Keywords:** circRNA, miRNA, droplet digital PCR, liquid biopsy, regional lymph node metastasis

## Abstract

**Simple Summary:**

Circular RNAs (circRNAs) consist of covalently closed structures without a free 3′ poly(A) tail or 5′ cap. Due to their loop structures, circRNAs are largely stable and resistant to RNA degradation by endonucleases. Therefore, they have much longer circulatory half-lives compared with linear RNAs, such as microRNA and long non-coding RNA. CircRNAs have multiple microRNA binding sites, and circRNA binding leads to regulation of the expression of those miRNAs and their target genes by acting as a competing endogenous RNA. However, the underlying molecular mechanisms and clinical correlation between circRNAs and tumor progression are not well understood. In the present study, we identified circRNA-102450 as a novel tumor suppressor and potential biomarker of regional lymph node metastasis from OSCC using liquid biopsy-based droplet digital PCR, a highly sensitive method for absolute quantification.

**Abstract:**

Circular RNAs (circRNAs), which form as covalently closed loop structures, have several biological functions such as regulation of cellular behavior by adsorbing microRNAs. However, there is limited information of circRNAs in oral squamous cell carcinoma (OSCC). Here, we aimed to elucidate the roles of aberrantly expressed circRNAs in OSCC. CircRNA microarray showed that circRNA-102450 was down-regulated in OSCC cells. Clinical validation of circRNA-102450 was performed using highly sensitive droplet digital PCR in preoperative liquid biopsy samples from 30 OSCC patients. Interestingly, none of 16 studied patients with high circRNA-102450 had regional lymph node metastasis (RLNM), whereas 4 of 14 studied patients (28.5%) with low expression had pathologically proven RLNM. Overexpressed circRNA-102450 significantly inhibited the tumor metastatic properties of cell proliferation, migration, and invasion. Furthermore, circRNA-102450 directly bound to, and consequently down-regulated, miR-1178 in OSCC cells. Taken together, circRNA-102450 has a tumor suppressive effect via the circRNA-102450/miR-1178 axis and may be a novel potential marker of RLNM in OSCC patients.

## 1. Introduction

Oral squamous cell carcinoma (OSCC), the eighth most common cancer, is associated with a poor prognosis and high recurrence rate [1,2]. Although research on OSCC has progressed substantially, surgery remains the only potentially curative treatment, and its five-year overall survival rate remains under 50% [3]. Therefore, understanding OSCC progression and metastasis at the molecular level is necessary to discover novel diagnostic and therapeutic targets for this disease.

Circular RNAs (circRNAs) consist of covalently closed structures without a free 3′ poly(A) tail or 5′ cap [4,5,6]. Due to their loop structures, circRNAs are largely stable and resistant to RNA degradation by endonucleases. Therefore, they have much longer circulatory half-lives compared with linear RNAs F.

Using advanced bioinformatics analyses, numerous circRNAs have been identified recently [7]. Jeck et al. reported that putative functions of circRNA are not only miRNA sponge but also regulation of transcription, interactions with RNA binding proteins, and translation of circRNAs [8]. Among them, circRNAs have multiple microRNA (miRNA) binding sites, and circRNA binding leads to regulation of the expression of those miRNAs and their target genes by acting as a competing endogenous RNA [4,9,10]. However, the underlying molecular mechanisms and clinical correlation between circRNAs and tumor progression are not well understood. Liquid biopsy (LB), a simple non-invasive test conducted in bodily fluids, allows repeated and convenient analysis of circulating nucleic acids, such as miRNAs and circRNAs, in cancer patients [11]. Therefore, a suitable approach for quantifying low levels of nucleic acids is required. Reverse-transcription (RT) droplet digital PCR (ddPCR) is a relatively new method of nucleic acid detection that involves partitioning a PCR solution into individual droplets to perform independent reactions, thus ensuring single-molecule sensitivity [12].

In the current study, we identified circRNA-102450 as a novel tumor suppressor and potential biomarker of regional lymph node metastasis (RLNM) from OSCC using LB-based RT-ddPCR.

## 2. Materials and Methods

### 2.1. Cells

Three human OSCC-derived cell lines (HSC-3, Sa3, and SAS) were purchased from RIKEN BioResource Center (Tsukuba, Japan) and the Japanese Collection of Research Bioresources Cell Bank (Ibaraki, Japan). We obtained human normal oral keratinocytes (HNOKs) from young healthy patients and cultured the cells as described previously [13,14,15,16]. All cells were used within seven passages after thawing.

### 2.2. RNA Extraction

Total RNA was isolated using Trizol reagent (Invitrogen, Carlsbad, CA, USA) according to the manufacturer’s instructions. For circRNA microarray analysis, total RNA (5 μg) was treated with RNase R (Lucigen, Middleton, WI, USA) at 37 °C for 15  min to degrade linear RNA.

### 2.3. circRNA Microarray Analysis

A circRNA microarray (Arraystar, Rockville, MD, USA) was used to investigate the differentially expressed circRNAs in three OSCC cell lines (HSC-3, Sa3, and SAS) and HNOKs. The sample preparation and microarray hybridization were performed according to the manufacturers’ protocols (Arraystar). Briefly, circRNAs were amplified and transcribed into fluorescent circRNAs using a random priming method. The labeled circRNAs were hybridized to the Arraystar Human circRNA Microarray (8 × 15 K). Scanned images were then imported to GenePix Pro 6.0 software (Axon, San Jose, CA, USA). The R software package (R version 3.1.2, R Foundation for Statistical Computing, Vienna, Austria) was used for quantitative normalization and subsequent data processing. circRNAs with statistically significant differential expression between two groups were identified by fold change filtering or scatter plot filtering. Hierarchical clustering was performed to identify distinguishable circRNA expression patterns among the samples. CircRNAs with a fold change in expression >3.0 were identified. All primary data from the microarray analysis have been uploaded to the Gene Expression Omnibus with accession number GSE145608.

### 2.4. RT-ddPCR Procedure

RT-ddPCR was performed using the One-Step RT-ddPCR Kit (Bio-Rad, Hercules, CA, USA). The 20 μL reaction mixture contained 4 μL RNA, 10 μL 2× One-Step RT-ddPCR Supermix (Bio-Rad), 0.8 μL 25 mmol/L manganese acetate solution (Bio-Rad), and the appropriate primer and probe at final concentrations of 0.2 and 0.16 μmol/L, respectively. The sequences of the divergent primers targeting circRNA-102450 were CACTCGCCCAAGTTTACCTG (forward) and ACTGAAAATGGCTTCGTTGATG (reverse), and that of the probe was TCCAAAATACTTTGTCCTGTGAGGCAGC. The reaction mixture was then prepared using 70 μL droplet generation oil and a droplet generator (Bio-Rad). The droplets were transferred to a 96-well plate, heat sealed with foil, and amplified on a conventional real-time PCR machine (Bio-Rad). The thermal profile consisted of 60 °C for 30 min, 95 °C for 10 min, followed by 40 cycles of 94 °C for 30 s and 60 °C for 1 min, and then a final 98 °C for 10 min. After PCR amplification, the plate was loaded onto the QX200 droplet reader (Bio-Rad) for data acquisition using QuantaSoft software (version 1.7, Bio-Rad). CircRNA-102450 was down-regulated in OSCC cells (HSC-3, Sa3, and SAS) compared with HNOKs. Among them, dramatical decrease of circRNA-102450 expressions were observed in HSC-3 and Sa3. Therefore, we had used HSC-3 and Sa3 for further experiments.

### 2.5. Preparation of LB Samples from OSCC Patients

Plasma samples were collected prior to surgery from 30 randomly selected OSCC patients without clinical RLNM (age, 33–84; average age, 65; 14 males and 16 females). All plasma samples were centrifuged for 10 min at 1600× *g*; the supernatants were carefully removed and transferred to a new tube followed by centrifugation again at 16,000× *g* for 10 min to remove residual blood cells. The circRNA level in plasma was measured by RT-ddPCR. The TNM stage has been assessed on enrolled surgical resected samples by two pathologists blinded to the patient’s clinical status.

### 2.6. miRNA Expression Analysis

PCR quantification of miRNAs was performed by TaqMan miRNA qPCR assays (Thermo Fisher Scientific, Waltham, MA, USA) after cDNA synthesis from 10 ng total RNA using the TaqMan MicroRNA Reverse Transcription Kit (Thermo Fisher Scientific). The expression levels of the miRNAs were normalized to those of U6.

### 2.7. Transfection of a circRNA-102450 Overexpression Vector

The circRNA-102450 sequence was cloned into pcDNA3.1 (+) (Thermo Fisher Scientific) to construct an overexpression vector (oe-circRNA-102450). pcDNA3.1 (+) without the target circRNA sequence was used as a negative control (Mock). The vectors were transfected into OSCC cells after confirmation of the sequences. Transfection assays were performed using Lipofectamine 3000 (Invitrogen) according to the manufacturer’s protocol. Cells were selected using 2 μg/mL puromycin (Santa Cruz Biotechnology, Santa Cruz, CA, USA), and the surviving cells were used as stable transfectants.

### 2.8. Cell Proliferation Assay

To investigate the effect of circRNA-102450 overexpression on cell proliferation, viable oe-circRNA-102450 or Mock cells were seeded in 6-well plates at 1 × 10^4^/well. Proliferation was assessed every 24 h for 120 h by trypsinizing the cells and counting in triplicate using a hemocytometer.

### 2.9. Cell Migration Assay

To evaluate the effect of circRNA-102450 overexpression on cell migration, we performed a wound healing assay. After uniform wounds were created using a micropipette among confluent cultures of oe-circRNA-102450 and Mock cells, the extent of wound closure was monitored visually every 12 h. The wound area was measured using Lenaraf 220 software (version 2.20b, http://www.vector.co.jp/soft/dl/win95/art/se312811.html, accessed on 29 July 2020), and the mean value obtained from three separate chambers was calculated [17].

### 2.10. Cell Invasion Assay

To evaluate the effect of circRNA-102450 overexpression on cell invasiveness, we performed an invasion assay. A total of 2.5 × 10^5^ cells were seeded on a polyethylene terephthalate membrane insert with a pore size of 8 μm in a Transwell apparatus (Becton–Dickinson Labware, Franklin Lakes, NJ, USA). In the lower chamber, 1 mL DMEM (Sigma-Aldrich, St. Louis, MO, USA) supplemented with 10% FBS (Sigma-Aldrich) was added. After the cells were incubated for 48 h at 37 °C, the insert was washed with PBS, and cells on the top surface of the insert were removed with a cotton swab. Cells adhering to the lower surface of the membrane were fixed with methanol and stained with crystal violet. The numbers of cells invading the pores in five random fields were counted using a light microscope at 100× magnification [18].

### 2.11. Luciferase Reporter Assay

The pMIR-REPORT luciferase reporter plasmid (Thermo Fisher Scientific) was constructed with or without the circRNA-102450 sequence. This reporter vector was co-transfected with an miR-1178 mimic or negative control mimic into OSCC cells, followed by a 48-h incubation. Luciferase activity was measured using a luciferase reporter assay system (Promega, Madison, WI, USA).

### 2.12. Statistical Analysis

Results are expressed as means ± standard deviation. Statistical analyses were performed using BellCurve for Excel. Differences between paired groups were analyzed by Student’s *t*-tests. *p* < 0.05 was considered to indicate statistical significance.

## 3. Results

### 3.1. circRNA Microarray Analysis in OSCC Cells

CircRNA microarray analysis was performed to evaluate the differential expression of circRNAs between OSCC cells and HNOKs. A scatter plot was used to assess the variations in circRNA expression between the cell types (Figure 1A). Unsupervised hierarchical clustering of circRNA expression patterns clearly identified the up- and down-regulated circRNAs in OSCC cells compared with HNOKs (Figure 1B). A total of 196 differentially expressed circRNAs with an average fold change >3.0 from three OSCC cells were identified, of which 41 were up-regulated and 155 down-regulated. We found that seven circRNAs were down-regulated by microarray analysis in all OSCC cells examined (Figure 1B). Of them, circRNA-102450 was selected for further studies in vitro and in vivo, since circRNA-102450 has several binding sites of cancer-related miRNAs.

### 3.2. Detection of circRNA-102450 by RT-ddPCR

The cDNA product of circRNAs obtained by the RT reaction using a specific primer is unsuitable for quantitative analysis of circRNA expression because it has multiple binding sites for PCR primers (Figure 2A, upper panel). Recently, using the divergent primers for PCR analysis was standard method for circRNA expression. Thus, we performed RT-ddPCR using divergent primers and a probe designed to target the junction site of circular formation between exons 14 and 13 (Figure 2A, lower panel). The PCR solution was divided randomly into water-in-oil droplets, and RT and PCR were conducted in each individual droplet. RT-ddPCR using a turn-on fluorescence probe could proceed only in the droplets containing circRNA-102450 and HPRP1 (an internal control gene) and these were identified as the fluorescence-positive droplets. The droplets without any target circRNA sequences were negative for fluorescence. The positive and negative droplets were counted for precise quantification of the circRNA (Figure 2B). RNase R is an exoribonuclease that degrades linear RNAs, but not circRNAs. After treatment with RNase R, the linear HPRP1 transcripts were not detected by RT-ddPCR, whereas circRNA-102450 was resistant to RNase R treatment and thus detected (Figure 2C). Several groups have reported circRNA expression data using conventional RT-PCR after treatment with RNase R [19,20,21], but as a consequence, they were unable to use internal control genes for PCR analysis. In the current study, by using divergent primers and the probe targeting the junction site, we achieved quantification of circRNA-102450 expression without having to use RNase R.

### 3.3. Expression of circRNA-102450 in OSCC Cells

We investigated circRNA-102450 expression by RT-ddPCR. The expression level of circRNA-102450 was considerably lower in the OSCC cell lines (HSC-3 and Sa3) than in the HNOKs (Figure 3). Similar to the data of circRNA-102450 expression, low density lipoprotein receptor, the cognate linear RNA of circRNA-102450, was down-regulated in OSCC cells compared with HNOKs by RT-qPCR.

### 3.4. Investigation of circRNA-102450 Expression in LB Samples

CircRNAs in LB plasma samples from OSCC patients were accurately quantified by RT-ddPCR (Figure 4A). The RT-ddPCR score for circRNA-102450 ranged from 0.017 to 0.19 (median, 0.05; Figure 4B). To determine the optimal cutoff RT-ddPCR score for circRNA-102450, we analyzed the scores from 30 patients using receiver operating characteristic (ROC) analysis. ROC analysis showed that the optimal cutoff score was 0.04 (area under the curve, 0.835; *p* < 0.05; Figure 4C). Patients with a score <0.04 were classified as the low circRNA-102450 group. In the analysis of clinical factors, the low circRNA-102450 group was closely associated with RLNM and TNM stage (*p* < 0.05; Figure 4D). In particular, the high circRNA-102450 group had no RLNM (0 in 16 cases), whereas low circRNA-102450 group showed pathologically proven RLNM (4 in 14 cases; 28.5%).

### 3.5. Inhibitory Effect of circRNA-102450 Overexpression on the Tumor Metastatic Properties of OSCC Cells

We investigated whether circRNA-102450 regulates OSCC progression. First, HSC-3 and Sa3 cells were transfected with the oe-circRNA-102450 vector to establish overexpression of circRNA-102450 (Figure 5). Then, cell proliferation, migration, and invasion assays were performed in these cells. We found a significant decrease in the proliferation of the oe-circRNA-102450 cells compared with the Mock cells (*p* < 0.05, Figure 6A). The migration assay showed that the wound area created by a micropipette was significantly smaller in the Mock cells compared with the oe-circRNA-102450 cells at 12 h (*p* < 0.05, Figure 6B). According to the invasion assay, the invasion ability was significantly lower in the oe-circRNA-102450 cells compared with the Mock cells at 24 h (*p* < 0.05, Figure 6C).

### 3.6. Expression of miR-1178 as a Target of circRNA-102450

Since circRNAs regulate cell behavior by adsorbing miRNAs [22,23], we investigated the potential miRNA targets of circRNA-102450 using bioinformatics analysis (Circular RNA Interactome, https://circinteractome.nia.nih.gov/, accessed on 29 January 2021). We identified a binding site for miR-1178 within the circRNA-102450 sequence. To verify that circRNA-102450 binds to miR-1178, a luciferase reporter assay was performed (Figure 7A). The luciferase activity of the circRNA-102450 reporter vector was reduced by miR-1178 transfection but not by transfection of the negative control mimic into OSCC cells (*p* < 0.05, Figure 7B). We then evaluated the endogenous expression level of miR-1178 in OSCC cells and found that miR-1178 was up-regulated in OSCC cells compared with HNOKs (*p* < 0.05, Figure 7C). Furthermore, miR-1178 was down-regulated in the oe-circRNA-102450 cells compared with the Mock cells (*p* < 0.05, Figure 7D). These data suggest that circRNA-102450 directly interacts with miR-1178 to regulate its expression negatively.

## 4. Discussion

Accumulating research has indicated that abnormal expression of circRNAs is linked to carcinogenesis and tumor progression in several cancer types [24]. Here, for the first time, we identified a novel OSCC-related circRNA, circRNA-102450, as a tumor suppressor using circRNA microarray and RT-ddPCR analyses. Numerous circRNAs have been reported to be either oncogenic or tumor suppressive factors. Oncogenic circRNAs are up-regulated in cancers, promote proliferation, and suppress apoptosis [25,26,27], whereas tumor suppressive circRNAs, as shown for circRNA-102450 in this study, are down-regulated and negatively correlated with cancer progression [28,29]. In OSCC, altered expression of circRNAs has been implicated in carcinogenesis and tumor progression [30,31]; however, the detailed functions and mechanisms of specific circRNAs remain poorly understood.

CircRNAs have several binding sites and act as sponges for miRNAs, which regulate the expression of target genes such as oncogenes and tumor suppressor genes [23]. Therefore, the circRNA/miRNA/mRNA axis has important roles in cancer development and progression [24,32]. Using bioinformatics databases of circRNAs, circRNA-102450 is predicted to target 15 potential miRNAs. In addition to our results on miR-1178, which is highly expressed in OSCC cells (Figure 7C), several studies indicate that miR-1178 acts as an oncogenic miRNA via the FAK signaling pathway in head and neck, bladder, and pancreatic cancers [5,33,34]. Consistent with the previous studies [35], it has been also demonstrated that E-cadherin expression, which is a recognized marker of invasiveness and metastasis, was increase in oe-circRNA-102450 cells (Appendix A).

LB of body fluids is commonly used for the diagnosis of multiple human diseases [36,37]. Although miRNAs have been widely assessed as biomarkers in body fluids [38,39], it was recently reported that circRNAs are highly resistant to RNA degradation by exonucleases compared with miRNAs [4,9,10]. CircRNAs are also widely distributed not only within cells but in the extracellular space and various body fluids [26,40]. Therefore, the high stability of circRNAs is beneficial for its use as a biomarker. A suitable approach for quantifying circRNAs detected in the limited amounts of LB samples is required. RT-ddPCR is an accurate method for detecting low levels of nucleic acids in plasma compared with conventional or real-time RT-PCR, which are associated with significant errors in circRNA quantification [12,26]. In the current study, to investigate circRNAs secreted in the plasma of OSCC patients, we applied highly sensitive and specific RT-ddPCR for absolute quantification. Using LB-based RT-ddPCR, the expression of circRNA-102450 was found to be related to the clinicopathological features of RLNM and TNM stage in OSCC patients. Interestingly, none of the 16 patients with high circRNA-102450 expression had RLNM (0 of 16 cases: 0%), whereas 4 of 14 patients (28.5%) with low expression had pathologically proven RLNM (Figure 4). RLNM of OSCC frequently occurs via the lymphatic system in the submandibular and cervical lymph nodes, even in the early cancer stages, and RLNM is the most important prognostic factor for OSCC patients [41,42,43,44]. Consequently, appropriate neck dissection is the standard treatment for OSCC patients. Therefore, in the future, collecting a large number of LB samples from the patients may help for neck dissection decision-making in OSCC patients who are clinically negative for RLNM.

The current study also demonstrated that circRNA-102450 significantly inhibits OSCC cell properties by directly binding to miR-1178 and subsequently reducing the tumor-suppressive capability of miR-1178. Thus, circRNA-102450 may be an encouraging molecular target for developing a therapeutic strategy to inhibit tumor progression. However, the concept of circRNA-based therapy depends on circRNA stability, differential expression in distinct organs, and specificity for a certain disease. Since circRNAs regulate a large number of genes via the circRNA/miRNA/mRNA axis, further investigations of the detailed functions of circRNA-102450 and miR-1178 are needed to understand its off-target effects and cancer-specific delivery system.

In summary, we demonstrated here that circRNA-102450 is a potential novel immunomarker predicting RLNM in OSCC patients. Furthermore, the relationship between five-year survival rates and circRNA-102450 levels would be critical information for development of novel therapeutic strategies for OSCC.

## 5. Conclusions

We demonstrated that circRNA-102450 is a potential novel immunomarker predicting RLNM in OSCC patients. Furthermore, our data suggest that circRNA-102450 might be an encouraging molecular target for development of novel therapeutic strategies for OSCC.

## Figures and Tables

**Figure 1 cancers-13-05708-f001:**
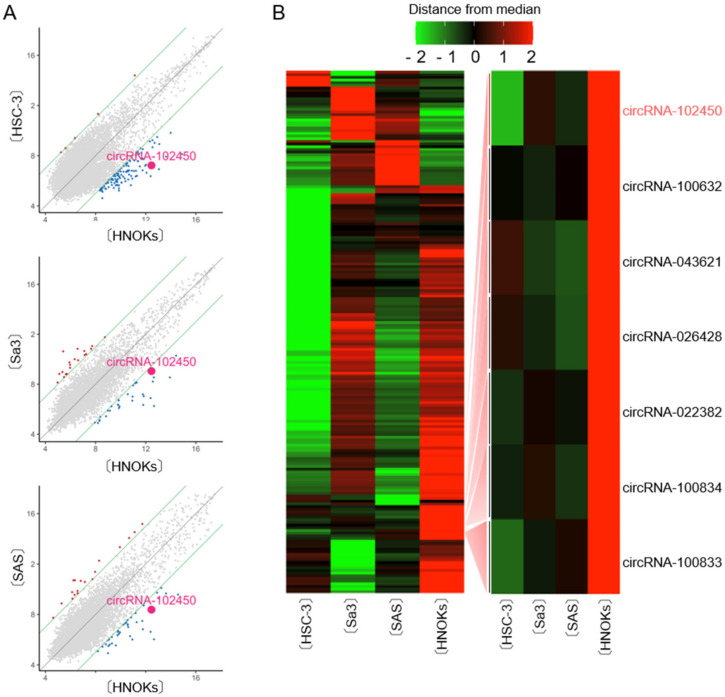
CircRNA microarray analysis in OSCC cells. (**A**) Scatter plot showing the variations in circRNA expression between the OSCC cell lines (HSC-3, Sa3, and SAS) and HNOKs. The circRNAs above the top green line and those below the bottom green line displayed a greater than 3.0-fold change in expression between the two samples compared. (**B**) Hierarchical clustering of circRNA expression profiles among the OSCC cells (HSC-3, Sa3, and SAS) and HNOKs. Red and green suggest high and low relative expression, respectively, in OSCC cells.

**Figure 2 cancers-13-05708-f002:**
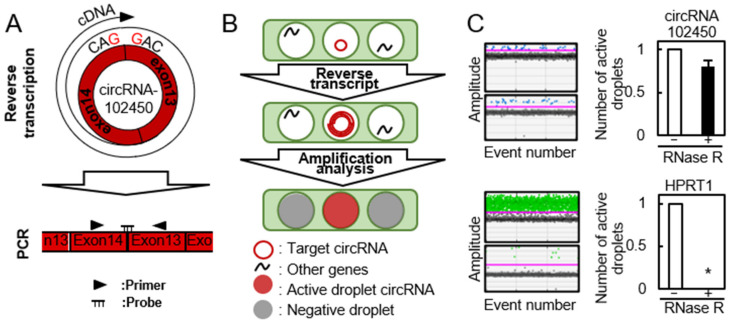
Detection of circRNA-102450 by RT-ddPCR. (**A**) Schematic illustration of the RT reaction and divergent primers and probe designed to target circRNA-102450. (**B**) Schematic illustration of the workflow for One-Step RT-ddPCR. (**C**) Expression of circRNA-102450 and HPRP1 after treatment with or without RNase R by RT-ddPCR. Data are expressed as means ± SD (* *p* < 0.05).

**Figure 3 cancers-13-05708-f003:**
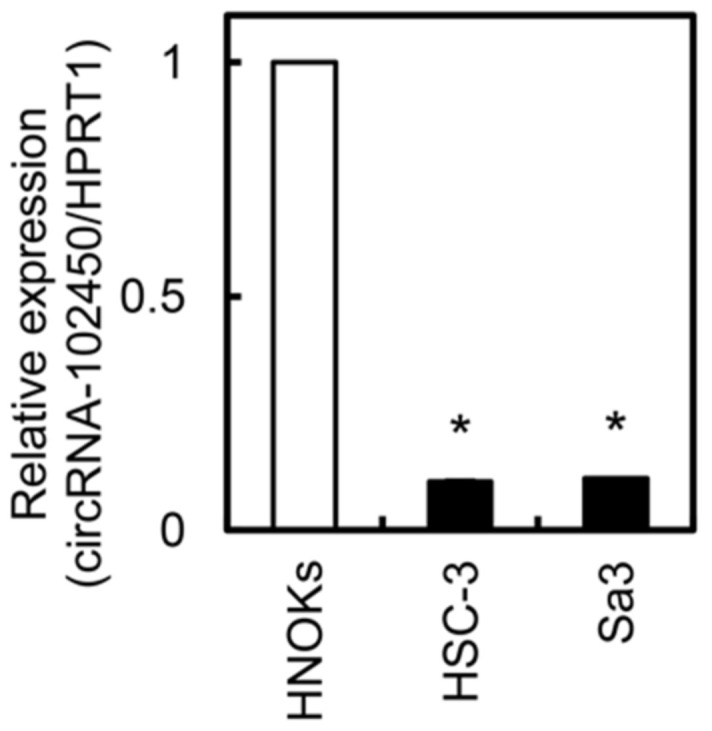
Expression of circRNA-102450 in OSCC cells. The expression of circRNA-102450 in OSCC cell lines (HSC-3 and Sa3) and HNOKs was evaluated by RT-ddPCR. Data are expressed as means ± SD (* *p* < 0.05).

**Figure 4 cancers-13-05708-f004:**
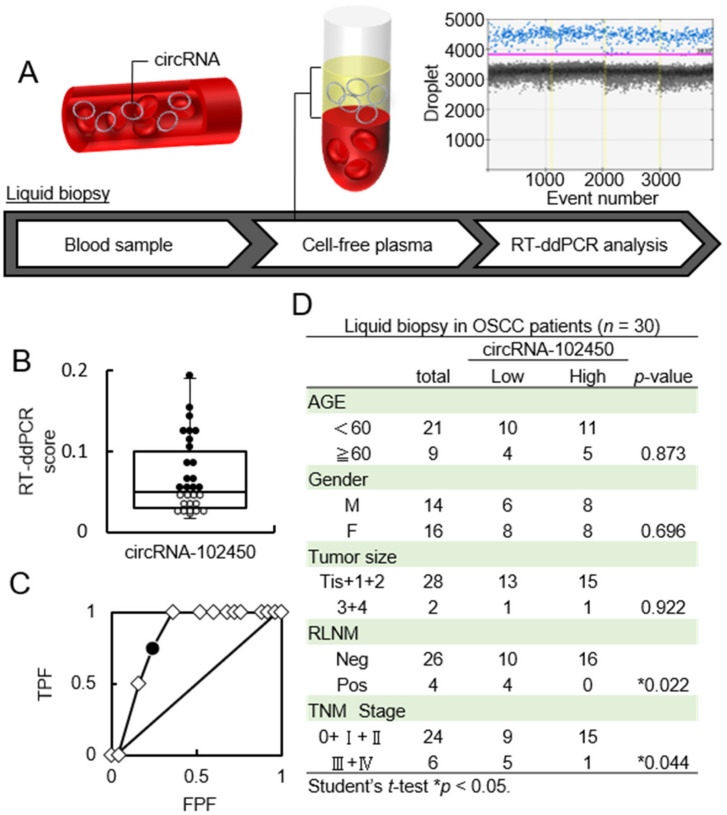
Investigation of the circRNA-102450 level in LB samples. (**A**) Workflow of the quantification of circRNA-102450 expression in LB samples from OSCC patients (*n* = 30). (**B**) The RT-ddPCR scores for circRNA-102450 in LB samples (● > The cutoff score, ◦ ≤ The cutoff score). The scores for circRNA-102450 ranged from 0.017 to 0.19 (median, 0.05). (**C**) ROC analysis of the circRNA-102450 score. The area under the ROC curve was 0.835, and the cutoff score was 0.04(● = The cutoff score, ◇ = Plot). (**D**) Relationships between clinical parameters and circRNA-102450 expression.

**Figure 5 cancers-13-05708-f005:**
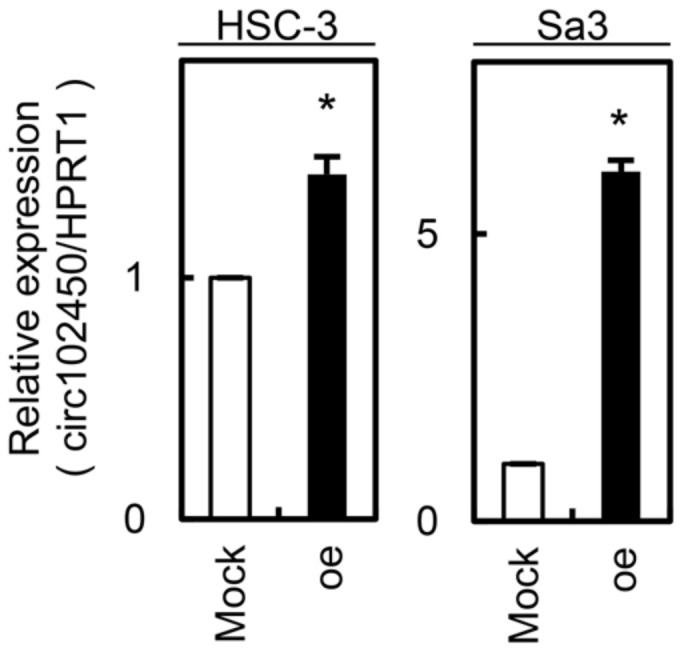
Establishment of circRNA-102450 overexpression in OSCC cells (HSC-3 and Sa3). Overexpression of circRNA-102450 was observed in HSC-3 and Sa3 cells after transfection with a circRNA-102450 overexpression vector. Data are expressed as means ± SD (* *p* < 0.05). oe, oe-circRNA-102450 cells.

**Figure 6 cancers-13-05708-f006:**
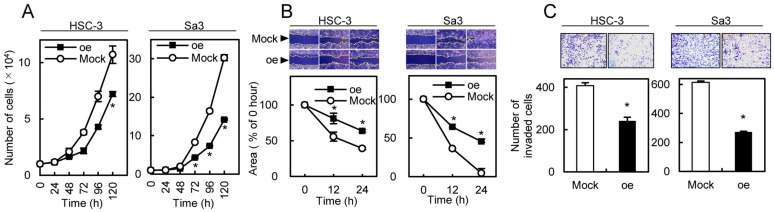
I Inhibitory effects of circRNA-102450 overexpression on the tumorigenic properties of OSCC cells. (**A**–**C**) The proliferation (**A**), migration (**B**), and invasion (**C**) of oe-circRNA-102450 and Mock cells (*n* = 3). Data are expressed as means ± SD from triplicate analyses of three independent assays (* *p* < 0.05). oe, oe-circRNA-102450 cells.

**Figure 7 cancers-13-05708-f007:**
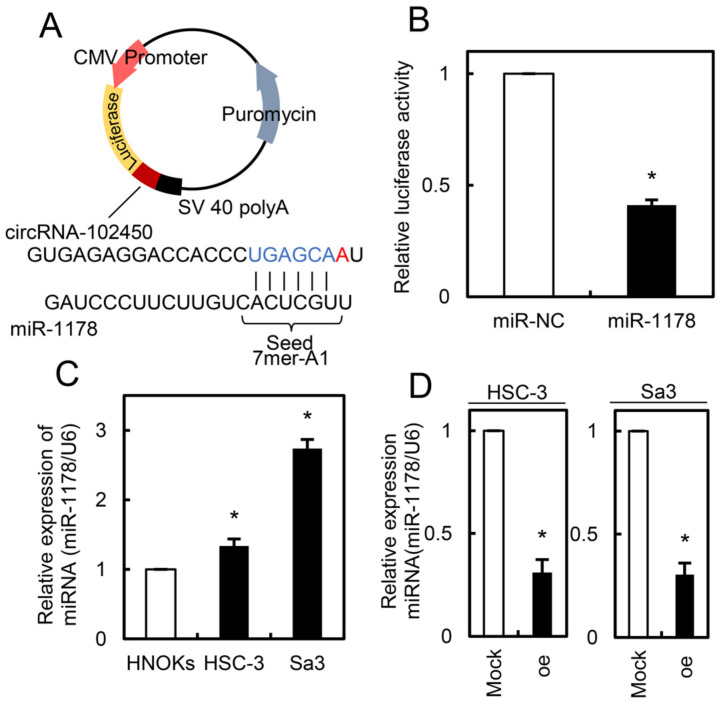
Expression of miR-1178 as a target of circRNA-102450. (**A**) Schematic illustration of the miR-1178 binding site in the circRNA-102450 sequence, inserted into a luciferase reporter vector. (**B**) Luciferase reporter assays showing that transfection of miR-1178 inhibited luciferase activity compared with transfection of the negative control miRNA. Data are expressed as means ± SD (* *p* < 0.05). (**C**) Up-regulation of miR-1178 in OSCC cells (HSC-3 and Sa3) compared with HNOKs. Data are expressed as means ± SD (* *p* < 0.05). (**D**) Down-regulation of miR-1178 in oe-circRNA-102450 cells compared with the Mock cells. Data are expressed as means ± SD (* *p* < 0.05). oe, oe-circRNA-102450 cells.

## Data Availability

Our expression data were deposited in the GEO database (accession number: GSE145608).

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
