# Peer review of "Tumor Suppressive Circular RNA-102450: Development of a Novel Diagnostic Procedure for Lymph Node Metastasis from Oral Cancer"

_cancers, 2021, doi:10.3390/cancers13225708_

Round 1
Reviewer 1 Report
The manuscript by Ando et al. has improved to some extent. The authors corrected some points raised by the reviewers and added more information. However, I still have some concerns.
1. The authors write now in the manuscript ‘Of these, seven circRNAs were down-regulated by RT-ddPCR analysis in all OSCC cells examined (Fig. 1B). In addition, circRNA-102450 was the only circRNA with significantly (P < 0.05) down-regulated expression in the OSCC cells.’ (page, 4; lines, 190-195).
It is still not clear to me whether RT-ddPCR was performed for all selected seven circRNAs as the authors claim above? If so, why do they not provide primer sequences for all 7 circRNAs and do not show the PCR results for all of them? Fig. 1 B is just a zoom-in of microarray results and not results of ddPCR as indicated in the sentences above.
2. I asked previously about the statistical analysis that was applied by the authors for identification of differentially expressed circRNAs. In response, the authors changed the criteria of selection from FC>3 and P<0.05 to only FC>3. Does it mean that no statistical analysis was performed at all?
Author Response
Responses to the Reviewer 1:
We thank the Reviewer 1 for his/her careful and comprehensive evaluation of our manuscript. We appreciate the comments and suggestions for improvement. We revised the manuscript as indicated below to address the points raised by the reviewer.
Review comments:
Q1.
The authors write now in the manuscript ‘Of these, seven circRNAs were down-regulated by RT-ddPCR analysis in all OSCC cells examined (Fig. 1B). In addition, circRNA-102450 was the only circRNA with significantly (P < 0.05) down-regulated expression in the OSCC cells.’ (page, 4; lines, 190-195).
It is still not clear to me whether RT-ddPCR was performed for all selected seven circRNAs as the authors claim above? If so, why do they not provide primer sequences for all 7 circRNAs and do not show the PCR results for all of them? Fig. 1 B is just a zoom-in of microarray results and not results of ddPCR as indicated in the sentences above.
Response (Q1):
We totally agree with the reviewer’s comments. As we are now carrying out the different project using other six circRNAs, we had focused on only circRNA-102450 in the present study and shown only names of six circRNAs.
Q2.
I asked previously about the statistical analysis that was applied by the authors for identification of differentially expressed circRNAs. In response, the authors changed the criteria of selection from FC>3 and P<0.05 to only FC>3. Does it mean that no statistical analysis was performed at all?
Responses (Q2):
Thank you for raising this point. Since we had performed microarray analysis among three OSCC cell lines and HNOKs (N=1), the selected circRNAs were identified using FC>3 criteria not statistical analysis. Therefore, we have deleted the sentence and made updates to the Materials and Methods section.
From
‘CircRNAs with a fold change in expression > 3.0 were identified as significantly differentially expressed.’
To
‘CircRNAs with a fold change in expression > 3.0 were identified.’ (page, 3; lines, 96)
Reviewer 2 Report
Dear Editor,
the paper of Ando T. et al. entitled Tumor Suppressive Circular RNA-102450: Development of a Novel
Diagnostic Procedure for Lymph Node Metastasis from Oral Cancer has been revised by authors;
nonetheless, for the publication of the final version in Cancers it is recommended to include the following
few revisions.
Minor
Page 9, Lines 324-326: The manuscript in the present form reports the Real-Time PCR and Western Blot
analysis performed to investigate E-cadherin expression in oe-circRNA-102450 cells. Nonetheless, it is
suggested to the authors specify the reason why also E-cadherin expression has been evaluated. Authors
may indicate it as follow: “Consistent with previous studies (Reference 35), it has been also demonstrated
that E-cadherin expression, which is a recognised marker of invasiveness and metastasis, was increase in oecircRNA-
102450 cells (Supplemental Figure 1).”
Page 3, Lines 125-127: The authors may improve the sentence “TNM stage was determined using surgical
resected samples. Two independent pathologists, neither of whom had knowledge of the patients’ clinical
status, made judgments of TNM stage” as follow: “TNM stage has been assessed on enrolled surgical resected
samples by two pathologists blinded to patient’s clinical status.”

Author Response
Responses to the Reviewer 2:
We thank the Reviewer 2 for his/her careful and comprehensive evaluation of our manuscript. We appreciate the comments and suggestions for improvement. We revised the manuscript as indicated below to address the points raised by the reviewer.
Review comments:
Q1.
Page 9, Lines 324-326: The manuscript in the present form reports the Real-Time PCR and Western Blot analysis performed to investigate E-cadherin expression in oe-circRNA-102450 cells. Nonetheless, it is suggested to the authors specify the reason why also E-cadherin expression has been evaluated. Authors may indicate it as follow: “Consistent with previous studies (Reference 35), it has been also demonstrated that E-cadherin expression, which is a recognised marker of invasiveness and metastasis, was increase in oecircRNA-102450 cells (Supplemental Figure 1).”
Responses (Q1):
We appreciate the reviewer’s comment. According to the reviewer’s suggestion, we have changed the comments in the Discussion section as follows:
From
‘Consistent with previous studies [35], we also demonstrated that E-cadherin expression was increased in oe-circRNA-102450 cells (Supplemental Figure 1).’
To
‘Consistent with previous studies [35], it has been also demonstrated that E-cadherin expression, which is a recognized marker of invasiveness and metastasis, was increase in oecircRNA-102450 cells (Supplemental Figure 1).’ (page, 9-10; lines, 326-329)
Q2.
Page 3, Lines 125-127: The authors may improve the sentence “TNM stage was determined using surgical resected samples. Two independent pathologists, neither of whom had knowledge of the patients’ clinical status, made judgments of TNM stage” as follow: “TNM stage has been assessed on enrolled surgical resected samples by two pathologists blinded to patient’s clinical status.”
Responses (Q2):
We appreciate the reviewer’s comment. According to the reviewer’s suggestion, we have changed the comments in the Materials and Methods section as follows:
From
‘TNM stage was determined using surgical resected samples. Two independent pathologists, neither of whom had knowledge of the patients’ clinical status, made judgments of TNM stage’
To
‘TNM stage has been assessed on enrolled surgical resected samples by two pathologists blinded to patient’s clinical status.’ (page, 3; lines, 125-126)
Round 2
Reviewer 1 Report
1. In my opinion there are only two options – if qPCR results for the other 6 circRNAs have been published elsewhere, the current manuscript should refer to this. If not, the results and primers used for the analysis of 6 circRNAs need to be showed in this manuscript. Otherwise, the authors cannot mention those 6 circRNAs at all.
2. The authors still use the word ‘significantly’ relating to the analysis of microarray data in line 189.
